# Overcoming catastrophic forgetting through weight consolidation and long-term memory

## Abstract

Sequential learning of multiple tasks in artificial neural networks using gradient descent leads to catastrophic forgetting, whereby previously learned knowledge is erased during learning of new, disjoint knowledge. Here, we propose a new approach to sequential learning which leverages the recent discovery of adversarial examples. We use adversarial subspaces from previous tasks to enable learning of new tasks with less interference. We apply our method to sequentially learning to classify digits 0, 1, 2 (task 1), 4, 5, 6, (task 2), and 7, 8, 9 (task 3) in MNIST (disjoint MNIST task). We compare and combine our Adversarial Direction (AD) method with the recently proposed Elastic Weight Consolidation (EWC) method for sequential learning. We train each task for 20 epochs, which yields good initial performance (99.24% correct task 1 performance). After training task 2, and then task 3, both plain gradient descent (PGD) and EWC largely forget task 1 (task 1 accuracy 32.95% for PGD and 41.02% for EWC), while our combined approach (AD+EWC) still achieves 94.53% correct on task 1. We obtain similar results with a much more difficult disjoint CIFAR10 task (70.10% initial task 1 performance, 67.73% after learning tasks 2 and 3 for AD+EWC, while PGD and EWC both fall to chance level). We confirm qualitatively similar results for EMNIST with 5 tasks and under 3 variants of our approach. Our results suggest that AD+EWC can provide better sequential learning performance than either PGD or EWC.

## 1 Introduction

Continual learning central to designing general A.I. systems that can learn new tasks sequentially without forgetting old tasks. However, current deep learning models based on stochastic gradient descent severely suffer from catastrophic forgetting (French, 1999; McCloskey & Cohen, 1989), in that they often forget all old tasks after training each new one. Inspired by the mammalian neocortex, which relies on processes of task-specific synaptic consolidation to enable continual learning (Cichon & Gan, 2015; Hayashi-Takagi et al., 2015; Yang et al., 2009; 2014), several concepts have been proposed, based on sequential Bayesian learning, which consist of applying a regularization function to a network trained by an old task to learn a new task. Many of these approaches work by finding a local minimum of the loss function for task B around the local region in the parameter space that was optimized for task A, such as learning without forgetting (Li & Hoiem, 2017), elastic weight consolidation (EWC) (Kirkpatrick et al., 2017), and incremental moment matching (Lee et al., 2017). Consider a model trained to perform task A: for input $\boldsymbol{X}_{taskA}$ and probability distribution $p(taskA)$, it produces outputs $f(\boldsymbol{X}_{taskA}, \boldsymbol{W}_{taskA}^*)$ by minimizing loss $L_{p(taskA)}(f(\boldsymbol{X}_{taskA}, \boldsymbol{W}))$. Training a second task B with input $\boldsymbol{X}_{taskB}$ and probability distribution $p(taskB)$ on the same network involves minimizing loss $L_{p(taskB)}(f(\boldsymbol{X}_{taskB}, \boldsymbol{W}_{taskA}^*))$. Previous approaches mentioned above attempt to restrict the parameters for task B to the local region around the optimum of task A, so as to minimally disturb what had been learned for task A. However, this might prevent the neural network from finding other regions in remote areas of the parameter space, which could contain a better local minimum of the loss function for the joint probability distribution of tasks A and B. To find this better local minimum, (Sprechmann et al., 2018) stores most of the data from old tasks into a working memory and replays it while training a new task. However, the working memory requires potentially large storage for the data from old tasks, and extra training time to replay that data.

Here, we introduce task-dependent memory units $\boldsymbol{M}_{task}$ and memory weights $\boldsymbol{W}_{task}$ to overcome catastrophic forgetting. Consider a two-task scenario, with an output function:

$$H(\boldsymbol{X}, \boldsymbol{M}_{task}, \boldsymbol{W}, \boldsymbol{W}_{task}) = f(\boldsymbol{X}, \boldsymbol{W}) + \mathbf{1}_{\text{taskA}}(g_{taskA}(\boldsymbol{M}_{taskA}, \boldsymbol{W}_{taskA})) + \mathbf{1}_{\text{taskB}}(g_{taskB}(\boldsymbol{M}_{taskB}, \boldsymbol{W}_{taskB})) \quad (1)$$

with input $\boldsymbol{X}$, task-dependent memory units $\boldsymbol{M}_{task}$, task-independent network weights $\boldsymbol{W}$, task-dependent memory weights $\boldsymbol{W}_{task}$, indicator function $\mathbf{1}_{\text{taskA}}()$, indicator function $\mathbf{1}_{\text{taskB}}()$, task-dependent function $g_{taskA}()$ and $g_{taskB}()$ and task-independent function $f()$. We update $\boldsymbol{W}$ in the EWC direction, update memory units $\boldsymbol{M}_{task}$ in an adversarial direction (further explained below), and update $\boldsymbol{W}_{task}$ in the gradient direction. Without any replaying of previous data, we can achieve a high accuracy on new tasks while minimally decreasing accuracy on old tasks. Our approach is not constrained to the local region around the parameters ($\boldsymbol{W}^*$) that are optimal for the old tasks. Instead, we create new parameter spaces ($\boldsymbol{W}^*, \boldsymbol{M}_{task}, \boldsymbol{W}_{task}$) that are good for both old and new tasks.

In the experimental results section, we show, using disjoint MNIST, disjoint EMNIST, and disjoint CIFAR10, that our Adversarial Memory Net (AMN) can be trained sequentially on multiple tasks while minimally decreasing accuracy on old tasks. Beyond these results which use a 5-layer fully connected network, we show that our approach can also apply to CNNs, although more research is necessary to enable EWC constraints to apply to convolutional and pool layers. We discuss complexity analysis, mathematical implications, and EWC in the discussion section.

## 2 BACKGROUND AND RELATED WORK

**Memory formation and retrieval in Hippocampus (HPC):** In the human brain, the hippocampus (Bakker et al., 2008) encodes detailed information in Cornu Ammonis 3 (CA3), which does pattern separation and transforms this information into abstract high-level information, then relayed to Cornu Ammonis 1 (CA1), which does pattern completion. During weight consolidation (Lesburguères et al., 2011; Squire & Alvarez, 1995; Frankland & Bontempi, 2005), the HPC fuses different features from different tasks into a coherent memory trace. Over days to weeks, as memories mature, HPC progressively stores permanent abstract high-level long-term memories to remote memory storage (neocortical areas). HPC can maintain and mediate their retrieval independently when the specific memory is in need.

**Adversarial examples, directions, subspaces, & programs, and long term memory:** Artificial neural networks are vulnerable to ***adversarial examples*** (Szegedy et al., 2013). By adding a carefully computed "noise" to an input picture, without changing the neural network, one can force the network into misclassification. The noise is usually computed by backpropagating the gradient in a so-called "***adversarial direction***" (Tramèr et al., 2017). Going to an adversarial direction, such as by using the fast gradient sign method (FGSD) (Goodfellow et al.), can help us generate adversarial examples that span a continuous subspace of large dimensionality (***adversarial subspace***). Because of "excessive linearity" in many neural networks (Tramèr et al., 2017; Goodfellow), due to features including Rectified linear units and Maxout, the adversarial subspace often takes a large portion of the total input space. Once an adversarial input lies in the adversarial subspace, nearby inputs also tend to lie in it.

Attack and defense researchers usually view adversarial examples as a curse of neural networks, but we view it as a gift to solve catastrophic forgetting. Input points that lie inside the adversarial subspace of each class lead the neural network to misclassify input images into that class. By extension, inputs which lie in the intersection of several adversarial subspaces that each belong to a different class may be misclassified into any of the underlying classes. Instead of adding input "noise" that lies inside the adversarial subspace calculated from other classes to force the network into misclassification, we can add "noise" calculated by the input's own correct class to assist correct classification. Our approach is to compute, in a task-dependent manner, this helpful "noise", and to store it into long-term memory units which are activated by different tasks. Thus, we propose that ***the intersection of the adversarial subspaces*** of all known classes is a representation of ***long term memory*** in neural network, although, we do not know how our brain represents permanent long-term memories. In essence, by storing task-dependent data into memory units, we add new dimensions to the original parameter space and create new spaces that are good for both old and new tasks.

Indeed, after we submit our paper to ArXiv for one month, a nice followup work (Elsayed et al., 2018) shows how a carefully computed ***adversarial program*** embedded in the input space can repurpose machine models to perform a new task without changing the parameters. This paper confirms that the

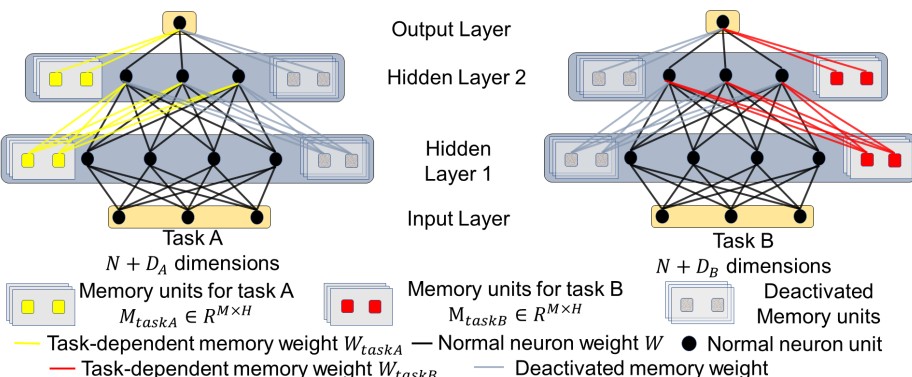

Figure 1: The structure of Adversarial Memory Network. N: dimensions of the normal neurons and normal weights (black), $D_A$: dimensions of the memory units and memory weights for task A (yellow),$D_B$: dimensions of the memory units and memory weights for task B (red), M: number of memory units, H: hidden dimension of a memory unit.

adversarial program (carefully computed noise), formed by finding the intersection of adversarial subspaces in our paper, is a good way to represent abstract information.

## 3 METHODS

In biological brains, during consolidation, abstract high-level memory is believed to be transferred from HPC to remote memory storage, such as the olfactory cortex(Lesburguères et al., 2011). Inspired by this idea, to achieve sequential learning with minimal interference, we add memory units into our network. We propose that one can designate and switch from task to task using dedicated task-dependent memory units. The intersection of the adversarial subspaces which represent the long-term memory in the network is stored in the memory units. The memory units, that store these long-term memory independently for each task, add additional dimension ($M_{task}, W_{task}$) to our old parameter spaces ($W$) and form new parameter spaces ($W, M_{task}, W_{task}$) that are good for both old and new tasks (Fig. 1 ).

### 3.1 EMBEDING ADVERSARIAL SUBSPACES INTO PARAMETER SPACE ($M_{task}, W_{task}, W$)

We want to embed the adversarial subspaces that have been found in the input space into our neural network, so that the parameters in our neural network behave like input images and span adversarial subspaces in the parameter space. Thus, we add task-dependent memory units (float tensors $M_{task}$, Fig. 1) and memory weights ($W_{task}$, Fig. 1) in each layer of the neural network. We update memory units in the adversarial direction. Each memory unit is a float tensor that can hold a float value. Memory units play a role similar to adversarial input images, capture the essence of each class, and span an intersection of adversarial subspaces in network parameter space.

### 3.2 ADVERSARIAL MEMORY NETWORK (AMN)

Task-dependent memory units ($M_{task} \in R^{M \times H}$, M is the number of memory units, H is the hidden size of a memory unit) store the intersection of adversarial subspaces in it which plays a role of long-term memory in our network, leading to the adversarial memory network structure (Fig. 1). During training and testing, the output function for 2 tasks is shown in Eqn. 1. Only memory units corresponding to the current task (yellow for task A or red for task B, chosen by indicator function in Eqn.1) and normal neurons (black) will be activated. The weights for memory units are called **memory weights** ($W_{task}$) and the weights for normal neurons are called **normal weights** ($W$).

Forward rules for each layer:
1. Deactivate the memory units not for the current task.
2. $Output = W_{task} * M_{task} + W * X_{normal\_neurons} + \textbf{bias}.$

What abstract long-term memory should we write into the memory units? The intersection of adversarial subspaces. How can we formulate it? We update memory units with gradients following an adversarial direction which can help us find the intersection of the adversarial subspaces in the parameter space. Thus, we have a joint abstract long-term memory of different classes available for each task in our neural network. In this work, we use the Fast Gradient Sign Method (FGSD) $\epsilon sign(\nabla_M L(M, y_{target}))$, where $M$ are memory units, $y_{target}$ is current input's true classes, as the adversarial direction to replace the gradient direction and use it to update memory units.

Backward rules for each layer:

$$O_{Gradients} = \begin{cases} \epsilon \ sign \ (O_{Gradients}) & if \ O \ is \ memory \ unit \\ O_{Gradients} & else \end{cases}$$

We use Softmax or Sigmoid cross entropy loss ( $L_{p(taskB)}(H(X, M_{task}, W, W_{task}))$ the first part in Eqn.2) to calculate the gradients for task-dependent memory units ($M_{task}$) and memory weights ($W_{task}$). To keep the normal weight close to the learned parameters of old tasks, we use total loss (Eqn. 2) with EWC constrains to calculate the gradients for normal weights ($W$). Consider a 2-task scenario, the loss function when we train task B is, where $F$ is the fisher information matrix, $\lambda$ sets how important the old task is compared to the new one:

$$Total \ Loss = L_{p(taskB)}(H(X, M_{task}, W, W_{task})) + \sum_i \lambda F_i (W_i - W^*_{taskA,i})^2 \quad (2)$$

## 4 EXPERIMENT RESULTS

We test our adversarial memory network with three datasets, 3 tasks for disjoint MINST tasks and disjoint CIFAR-10 tasks, 5 tasks for disjoint EMINST dataset (table 1). We train tasks sequentially, each for 20 epochs. We show that our approach can also apply to CNN (LeNet) by replacing the fully connected layers (FC) with our adversarial memory layers, and keeping the convolution and maxpool layers. We explore 5 different training methods:

1. EWC only: deactivate all the memory units so that our adversarial memory network becomes a fully connected network, and use elastic weight consolidation only.

2. AD: we use the adversarial memory units only. After finishing the training of task 1, we freeze all the normal weights, and only allow the update of memory units in adversarial direction and memory weights in gradient direction in the latter tasks.

3. EWC + AD: normal weights are updated in EWC direction. The memory units are updated in adversarial direction and memory weights are in gradient direction.

4. PGD: we deactivate all the memory units so that our adversarial memory network become a fully connected network, and use plain gradient descent.

5. One Image Storage (OIS): we use the same training method as EWC + AD, except for the following differences. In One Image Storage Big (OISB): the dimension of memory units is 1 x (28*28). The size is the same as storing a one-channel MNIST image in each layer. In One Image Storage Small (OISS): the dimension of memory units is 1 x (5*5).

We test 4 different networks (in table 2) with different hyperparameters. Note how the settings with 3 output neurons for MINST, CIFAR10, or with 5 outputs neurons for EMNIST (networks 1 & 2 in Table 2) will associate several labels with each output neuron, one per task. We use these settings for easy comparison with previous work, but we note that the settings with 9 output neurons for MINST, CIFAR10, or with 25 outputs neurons for EMNIST (networks 3 & 4) may be preferred in practice because they yield unambiguous classification results.

From Fig. 3 and Fig. 4, adversarial memory network has memory units to store the intersection of adversarial subspaces and plays a role as the abstract long-term memory. It uses EWC to form a joint probability distribution for sequential tasks represented by the normal memory (EWC + AD: blue, OISB: black, OISS: magenta curve). It outperforms the fully connected network with EWC (red curve) or plain gradient descent (PGD; yellow curve) by a large margin in both datasets and for our 4 hyperparameter sets. In the disjoint MINST tasks, EWC + AD achieves high accuracy and

Table 1: Disjoint MINST tasks, disjoint CIFAR10 tasks and disjoint EMNIST tasks. We train our Adversarial Memory Network sequentially on disjoint tasks for each dataset.

| Tasks / Datasets | Task1 | Task2 | Task3 | Task4 | Task5 |
|---|---|---|---|---|---|
| MNIST | Digits 0,1,2 | Digits 4,5,6 | Digits 7,8,9 | N/A | N/A |
| CIFAR10 | Airplane Automobile Bird | Deer Dog Frog | Horse Ship Truck | N/A | N/A |
| EMNIST | Letter A,B,C,D,E | Letter F,G,H,I,J | Letter K,L,M,N,O | Letter P,Q,R,S,T | Letter U,V,W,X,Y |

Table 2: Hyperparameters for 4 different network structures

| Hyperparameters | Network1 | Network2 | Network3 | Network4 |
|---|---|---|---|---|
| Number of hidden layer | 4 | 4 | 4 | 4 |
| Width of hidden layer | 300 | 300 | 300 | 300 |
| Width of Output layer(whether to have overlap output or not) | EMNIST: 5 MINST, CIFAR10: 3 | EMNIST: 5 MINST, CIFAR10: 3 | EMNIST: 25 MINST, CIFAR10: 9 | EMNIST: 25 MINST, CIFAR10: 9 |
| Loss function | Sigmoid cross entropy loss | SoftMax cross entropy loss | Sigmoid cross entropy loss | SoftMax cross entropy loss |
| Dropout | Yes | Yes | Yes | Yes |
| Memory units dimension in each layer | OISB: 1 x (28*28) OISS:1 x (5*5) | OISB: 1 x (28*28) OISS:1 x (5*5) | OISB: 1 x (28*28) OISS:1 x (5*5) | OISB: 1 x (28*28) OISS:1 x (5*5) |
| | EWC,AD,EWC+AD: 300 x 9 | EWC,AD,EWC+AD: 300 x 9 | EWC, AD, EWC+AD: 300 x 9 | EWC, AD, EWC+AD: 300 x 9 |
| Adversarial direction | Fast gradient sign method | Fast gradient sign method | Fast gradient sign method | Fast gradient sign method |
| Epsilon in FGSM | 0.2 | 0.2 | 0.2 | 0.2 |
| Optimizer | SGD + 0.15 momentum | SGD + 0.15 momentum | SGD + 0.15 momentum | SGD + 0.15 momentum |

barely forgets the old tasks at the same time. In the disjoint CIFAR10 tasks, we cannot achieve a high initial accuracy because we only use a fully connected structure and do not have a convolutional structure at all. But the overall result is the same as with the MINST tasks. The sigmoid cross entropy loss converges slower than the SoftMax cross entropy loss and has a lower initial accuracy than the SoftMax cross entropy loss in disjoint CIFAR10 tasks. However, from Fig. 4 c), we can see that, in more complicated tasks such as disjoint CIFAR10, where tasks do not share similar low-level features, the sigmoid function may be able to pass gradients only through the active tasks since the target vector is in one-hot expression. Thus, it gives us a better accuracy for task 1 after training task 2 and task 3. The AD alone (green curve) does not forget task 1 at all, because it is frozen. But the tradeoff is that we cannot achieve high accuracies in the latter tasks. In contrast, EWC+AD can learn the latter tasks very well while only minimally decreasing task 1 accuracy. The OISB and OISS methods perform slightly worse than AD+EWC, because they use fewer memory units and the first dimension of the memory unit tensors is one for the sake of scalability. You can view OISB as storing a size of a 1-channel MINST image per layer and OISS as storing a size of 1-channel 5*5 image per layer. Yet, both OISB and IOSS still perform much better than EWC and PGD. In Fig. 5 a), the overall result for 5 disjoint EMNIST tasks (5 letters per task) is the same with the MINST tasks.

Table 3: VGG16. Complexity analysis: parameter cost, memory cost in forward and backward pass. Duplicate a new VGG16, EWC + AD, OISB, OISS, EWC

| Complexity Analysis | VGG 16 | Duplicate a VGG16 | EWC + AD | OISB | OISS | EWC |
|---|---|---|---|---|---|---|
| Memory cost (forward and backward cost) | $96 * 2$ MB / image | $96 * 2$ MB / Image x # of task (100% increase) | $6.48 * 10^{-2} * 2$ MB / Image x # of tasks (0.0675 % increase) | $1.88 * 10^{-2} * 2$ MB / Image x # of tasks (0.0196% increase) | $6 * 10^{-4} * 2$ MB / Image x # of tasks (0.000625 increase) | 0 MB / Image x # of tasks (0% increase) |
| Parameter cost | 138 M | 138 M x # of tasks (100% increase) | 89.8641 M x # of tasks (65.12% increase) | 26.0939 M x # of tasks (18.91% increase) | 0.8321 M x # of tasks (0.6% increase) | 0 M x # of tasks (0% increase) |

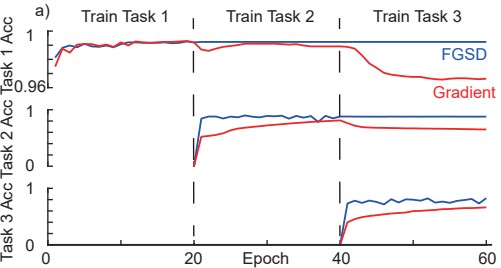 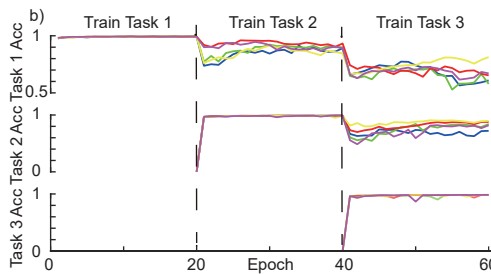

Figure 2: Disjoint MINST tasks in network 2. a) AD only. We update memory units with the gradient following the adversarial direction using the Fast Gradient Sign Method (blue curve). For comparison, we also update memory units using the normal gradient direction (red curve), but this does not work as well. b) AD+EWC. Varying the number hidden dimension of memory units with 300 memory units per layer. Blue curve: 300 x 1. Green curve: 300 x 3 . Red curve: 300 x 6. Yellow curve: 300 x 9. Magenta curve: 300 x 12.

In Fig. 2 a), we demonstrate that following an adversarial direction and finding the intersection of adversarial memory subspaces is crucial, by comparing updating the memory units in the gradient direction versus the adversarial direction (FGSD). The accuracy for adversarial memory units - FGSD (blue curve) is much higher than Gradient memory units (red curve) and does not vary with the number of epochs after being trained. As a result, we argue that the intersection of adversarial subspaces in memory units is how our neural network represents the abstract long-term information. If we store this information, we can view it as the long-term memory of our neural network. We view the memory units as storage of the essence of a lot of input pictures from previous tasks and it is ready to retrieve the corresponding one when we test on a specific task. From Fig. 2 b), while keep the number of memory units as 300 (equal to the number of neurons), by varying the hidden dimension of memory units in each task, we find that a low number hidden dimension of memory units (1 ~ 3) may not be sufficient to represent high-level memory information. Yet, too many hidden dimensions (above 9) cause too much disturbance, which also decreases accuracy. Best accuracy was obtained for 6 to 9 hidden dimensions in our experiments.

Many convolutional neural networks (CNN) have some fully connected layers (FC) at the output. These are used to classify the features generated by convolutional layers. To test our method with CNNs, we use the conv and pool layers from LeNet (LeCun et al., 1998) and replace its FC layers by adversarial memory layers (300 hidden units for each layer). We use Softmax cross entropy loss with overlapped 3 outputs. AD + EWC, OISB achieves high accuracy on new tasks, and achieves a better accuracy than PGD on remembering old tasks (Fig. 5 b)). One difficulty with applying AD+EWC to CNNs, which will need to be addressed in future research, is to better understand how to apply EWC constraints to convolutional and pool layers. Once we know how to constrain convolutional and pool layers, we may be able to fully adapt our adversarial memory network approach to a wide range of convolutional neural network structures.

## 5 DISCUSSION

**Complexity analysis: parameter cost, memory cost and extra training cost:** One perhaps obvious approach to avoiding interference between sequential tasks might be to use a separate network for each task. However, such method is not scalable with the number of tasks, because there is a one-to-one mapping from network to task and it requires to duplicate every neurons to build a new network. Although we also introduce task-dependent neurons in the FC structure, for our OISS method, we only introduce 25 extra neurons per task per layer compared to 300 neurons increase per task per layer if we use a separate network. The increase of memory cost is 25+25*300 (for 25 units plus the weights from those 25 units to the 300 normal neurons of the next layer) compared to 300*300 in a separate network (duplication a whole layer and its weights). In our CNN experiments, we show that changing the last 3 FC layers(classification layers) into our AMN and keep the conv and pool layers(feature extraction layers) in the original network to overcoming catastrophic forgetting.

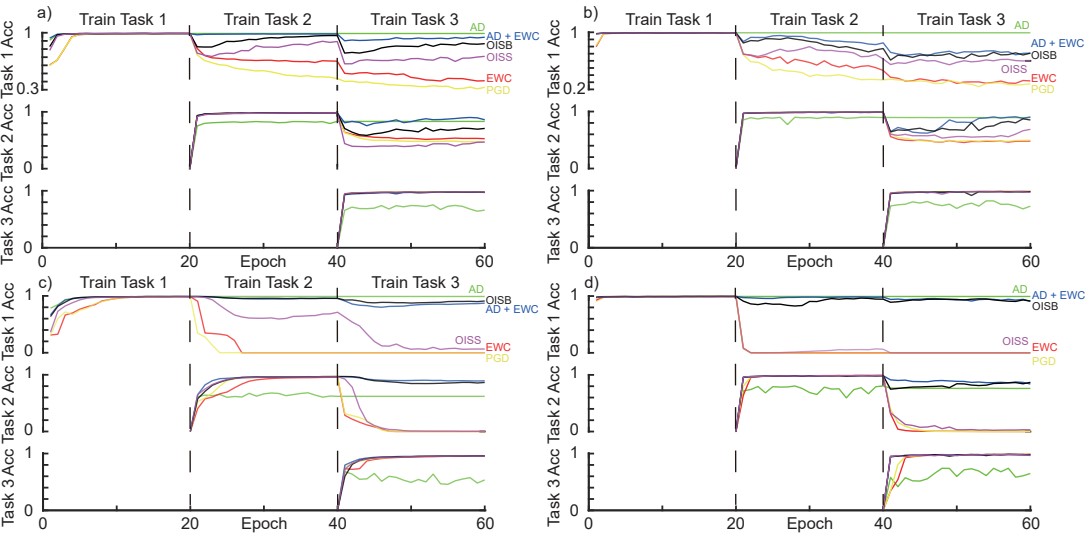

Figure 3: Disjoint MINST tasks. The red curve is using EWC alone (training method 1). The green curve is using adversarial memory units alone (training method 2). The blue curve is EWC and adversarial memory training (training method 3). The yellow curve is PGD (training method 4).The black curve is OISB and the magenta cur is OSSS (training method 5). Each subfigure is a) network 1, b) network 2, c) network 3, d) network 4, in Table 2.

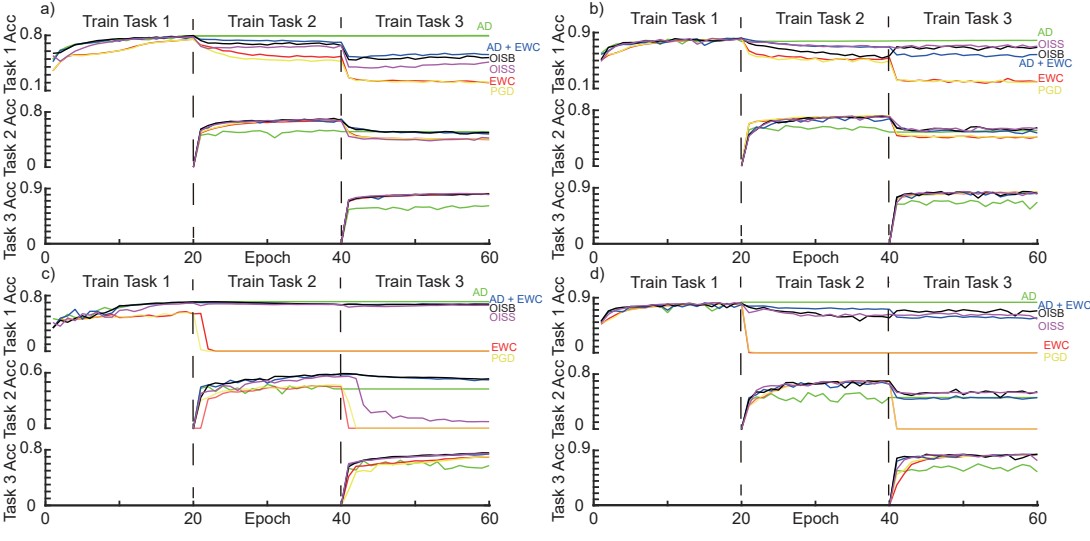

Figure 4: Disjoint CIFAR10 tasks. Organization is as in Fig. 3

Adding memory units to classification layers and restricting feature extraction layers to the local region of the old optimum is enough for us to remember old tasks. Consider a deep network such as VGG16 in table 3 ((Simonyan & Zisserman, 2014)), if we use a separate network the increase of parameter is 138 million parameters(100% increase)/task and (96*2) MB/(image,task) memory (100% increase) for forward and backward pass. However, for our OISS method we only introduce 0.832 million parameters (0.6% increase)/task and 1200 B/(image,task) memory for forward and backward (0.000625% increase). If we can find better adversarial direction than FGSM, we can use smaller memory units per task and still achieve the same accuracy. Another interesting research

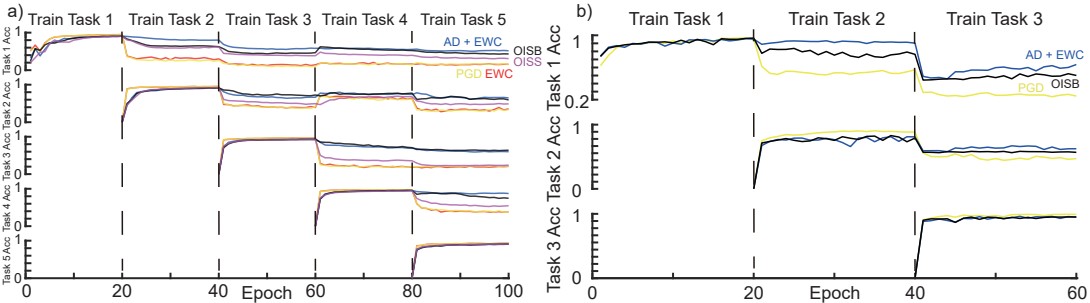

Figure 5: a) Disjoint EMNIST tasks. The red curve is using EWC alone (training method 1). The blue curve is EWC and adversarial memory training (training method 3). The yellow curve is PGD (training method 4). The black curve is OISB and the magenta curve is OSSS (training method 5). We use Sigmoid cross entropy loss with overlapped 5 outputs. b) Disjoint CIFAR10 tasks using a convolutional network. The convolutional and pool layers are the same as LeNet, but we replace the last 3 fully connected layers by our adversarial memory layers (300 neurons for each layer). After training task 1, we freeze the parameters in the convolutional layers. we use AD+EWC or OISB on adversarial memory layers (blue curve) . The yellow curve is the PGD method. We use Softmax cross entropy loss with overlapped 3 outputs.

direction is how to find the intersection of adversarial subspaces belonging to all classes from all tasks sequentially. This allows all tasks to share one task-independent memory units. Another advantage of our method is that, since we do not need to replay data from old tasks while training the new task, we save extra training cost. This gives us a good way to do online learning.

**Mathematical explanation:** Methods like EWC only apply a regularization function to a network trained by old tasks, to learn a new task based on sequential Bayesian inference, by finding a sub optimum local minima ($W^*_{B\ around\ A}$) by minimizing $L_{p(taskB)}(f(X_{taskB}, W^*_A))$ of task B around the local region of parameters space of task A ($W^*_A$). When the sequential tasks do not share similar low-level features, we usually cannot find a good joint probability distribution around the local region of parameters space of task A. Even worse is that the neural network fails to traverse other regions in the remote area of the space which might have a much better local minimum ($W^*_{A,B}$) by minimizing $L_{p(taskA,taskB)}(f(X_{taskA,taskB}, W))$ for the joint probability distribution of tasks A and B. Although we cannot revisit the data from earlier tasks in sequential learning, we can store some abstract long-term features shared by the data from earlier tasks into the memory units of our neural network and use this knowledge to classify the old test data from earlier tasks. In our adversarial memory network case, we store the adversarial gradient into memory units, which forms the intersection of adversarial subspaces inside the memory units $M_{task}$. When we test on each task, we combine the information stored in the specific memory units with our normal weights trained by the EWC to form new spaces ($W^*, M_{task}, W_{task}$). These new spaces usually give us a good estimate of the current test task, even though the normal weights have been modified by the EWC algorithm. This opens a new world for us that we build new curvature spaces which our experiments show have better local minima of the loss function for tasks A and B, by combination of normal weights, memory weights, and memory units.

**Comments on EWC:** In our experiments, EWC alone did not prevent catastrophic forgetting, though it did perform better than PGD. Yet, our EWC implementation works well with the original permuted MINST task used by the authors of EWC. Here, we tested networks with shared outputs and individual outputs separately, to evaluate the influence of having different classes from different tasks share the same label. With shared outputs, as used in the original EWC work except that our dataset is not permuted MNIST, after finishing training of task 3, task 1 accuracy for both EWC and PGD fell to chance level. With individual outputs, during the training of task 2 and task 3, both EWC and PGD rapidly decreased to 0% accuracy because the neural network cannot map task 1 to the corresponding correct outputs properly. In comparison, our EWC+AD, OISB, OISS and AD methods work well in both output scenarios.

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
