# OpenReview forum: "Overcoming catastrophic forgetting through weight consolidation and long-term memory"
_ICLR.cc/2019/Conference_

### Official Review · AnonReviewer1 · 2018-10-29
**Counter-intuitive adversarial memory units lacking persuading theoretical or empirical explanations**

**Rating:** 4
**Confidence:** 5

**Review:**

This paper proposes a new approach to sequential learning by introducing an adversarial memory unit for each new task and uses EWC as a regularizer for training other parts of the network on the new task.
The memory units are trained with Fast Gradient Sign Method to increase the loss, and they are connected to the next layer with weights trained to decrease the loss.
It shows superior performance than EWC and the plain gradient descent baseline on disjoint MNIST/CIFAR10 and EMNIST. The authors also share their experience with EWC, which provides useful feedbacks to the community.

The proposed adversarial memory unit is novel to the best of my knowledge. However, its motivation is not quite intuitive to me, and the authors fail to provide persuading explanations. My major concern is whether it is better to take the adversarial direction rather than the direction that decrease the loss for the memory units.

To support their ideas, the authors mentioned the paper "Adversarial Reprogramming of Neural Networks" and said this paper's "adversarial program" is formed by choosing the "intersection of adversarial subspaces" as in their paper. However, they (Elsayed et al. 2018) are actually finding such adversarial programs in the direction of decreasing the loss, which is contrary to finding the "intersection of adversarial subspaces".
The authors also want to support the pros of adversarial memory units by comparing against "Gradient" memory units that are trained to decrease the loss with the experiment shown in Figure 2. However, Figure 2a seems problematic to me, so I am not sure whether the authors are doing their experiments correctly. I think the experimental conditions for FGSD and Gradient are different, which makes the comparison meaningless. We can see that the network's accuracy with Adversarial memory unit on task 1 is a constant when the network is trained on task 2 and 3, because the network's weights (except memory units and their weights for task 2 and 3) and task 1's memory units are fixed, as described in the experimental setting for "AD". The accuracy on task 1 with Gradient memory units is changing when the network is trained on task 2 and 3, which means either the network's weights are changing or the memory unit is changing.

As a result, I don't think this paper will be accepted until the authors provide further explanations and results to support the adversarial memory unit, or clarify my misunderstandings in the comments above.

---

### Official Review · AnonReviewer3 · 2018-11-02
**Insufficient Experimental Validation**

**Rating:** 4
**Confidence:** 4

**Review:**

This paper proposes a novel continual learning method that stores intersection of adversarial subspaces into long-term memory units for each task, which is used used to characterize the given task at future tasks. This adversarial memory network requires supposedly less number of parameters for each task to store, compared to methods that stores explicit examples. The authors validated the proposed model on three datasets for continual learning, on which it obtains good performance when networks trained with plain gradient descent and elastic weight consolidation suffers from catastrophic forgetting.

Pros
- The idea of using adversarial subspaces to characterize a task is a novel idea which seems to work to some degree.

Cons

Experimental validation is lacking in many aspects.

- First, while the proposed method requires additional memory storage and parameters, it is not comparing against any of the existing work that increases network capacity or storing a small subset of the original dataset. To list a few that seems relevant, [Yoon et al. 18] proposes a network that can dynamically expand its capacity with minimal number of units per layer, and [Nguyen et al. 18] proposes to store a small subset called CoreSet that well-represent the task-specific dataset. To show that the proposed method is indeed effective in terms of accuracy over number of parameters, the authors should compare against such baselines with additional parameters. The increase in the network capacity reported in the paper seems quite large (over 60% for AD+EWC) and thus its effectiveness is questionable without such comparative study.

- Their implementation of EWC seems suboptimal as it is only applied to fully connected layers, and thus the EWC baseline performs much poorly than what are reported in many of the previous work, and performs comparable to PGD. Since EWC baseline is crippled the only message that is remaining is that the proposed method works better than simple PGD.

- The reported results using the proposed method shows some performance degradation on earlier tasks, which seems large considering the difficulty of the tasks. Again, the authors should compare against recent methods on continual learning so that the readers can understand how good these reported performances are.

- It is difficult to understand why storing adversarial subspaces helps, since there is no visualization or illustrations that provide intuitive explanations.

In sum, while the proposed model seems novel, its motivation is unclear and it is difficult to assess the effectiveness of the proposed method due to lack of experimental validation. Thus I recommend the rating of reject for this paper, until the authors provide additional experimental results for proper assessment of the method's effectiveness.

---

### Official Review · AnonReviewer2 · 2018-11-02
**Paper with Interesting novel ideas, but it needs major presentation improvements**

**Rating:** 4
**Confidence:** 4

**Review:**

The paper is about a new method for training neural networks in the continual learning setting, where tasks are presented in a sequential manner (and data from the previous task cannot be revisited). The method proposes a new architecture that adds task-parameters parameters to prevent catastrophic forgetting.

To my understanding, the paper proposes a modification to EWC in which the capacity of the network is augmented after a new task is added. Unlike similar methods (like Progressive networks, see bellow), this augmentation is input agnostic. It acts as a correction of the model parameters such that the new task can be easier to train while still maintaining the 'normal parameters' close to the ones of the initial task (as in EWC). I find this idea interesting and certainly worth publishing. In my view, the paper cannot be published in its current state. With the current presentation it is very difficult to understand what is being proposed. Please correct me if I misunderstood the work.

The writing of the manuscript needs significant improvement. I read it carefully several times and I am still not sure of how exactly the model is trained. I had to read the paper by Elsayed et al 2018, to have an idea of what could have been proposed here. As I mentioned, the paper has novel and interesting ideas, but it would be greatly improved with some important re-writing. Please find bellow some questions.

- In the second to last paragraph of page two, the authors say that: instead of adding a perturbation that would force the network to perform a misclassification, tune it using "the input's own correct class to assist correct classification". If the gradients are computed with respect to the correct class of a given input, why is this called an adversarial perturbation?

- Elsayed maintain the parameters of the first task fixed and train a fixed input-agnostic correction that can be added to the input such that a second task can be trained (with a re-mapping the outputs). Applying Elsayed et al 2018 to the continual learning setting, the model should only learn correction for task 2 (and 3). How do the authors compute the corrections for task 1? Computing a correction requires having access to the training data.

- The authors use the FGSM method to compute "adversarial perturbations". This method was proposed as a proxy for performing gradient descent to minimize the computational load required for finding adversarial examples. In this application, unlike the case of adversarial perturbations, the memories don't need to be constrained to be smaller than a given epsilon. What is the motivation of using this method? How do you explain the difference in the results.

- Having mentioned this, both W_task and M_task are updated by minimizing the same loss function (ignoring the difference of using FGSD or not). In that case, why is it needed to have a factorized form W_task * M_task instead of a single bias?

- Throughout the paper the authors say that the long term memory lies on the "intersection of adversarial subspaces". It is not clear at all why this should be the case. The authors do not explain adversarial subspaces corresponding to which model.

- The authors should cite the Progressive networks as this is a very related work. Unlike progressive networks, this work proposes and augmentation that is input agnostic which is interesting. https://arxiv.org/abs/1606.04671

- With EWC, once the model is trained, one does not need to know the task being evaluated at test time. This is not the case in the proposed model. This should be clarified. Also, when having many tasks mapping to the same input, the fair way of comparing to EWC would to have a different head per task. This baselines should be included.

- What are the task specific functions g_taskA and g_taskB?

- Adding an explicit algorithm, the exact loss functions used should help clarifying the proposed method.

- The paper would be stronger if more complex tasks would be added.

---

### Public Comment · ~Min_Lin1 · 2018-09-29
**Comments on the Experiments**

The experiments comparing with EWC could be not very fair in my opinion.
In literature, there are many different assumptions while performing continual learning,
The most difficult task setting would be assuming no knowledge of task boundary both during training and testing.
As a compromise, one can assume the boundary is known during training but unknown during testing, like in EWC.
Some of the works assume the task boundary is known during both training and testing, like in this paper, since there are task neurons, I assume during test the task id is required.

In my opinion, comparing two methods with different assumptions on the knowledge of task boundary is not fair.

---

> ### Author Response · Authors · 2018-09-29
> **EWC needs task ID during testing in the shared outputs case**
>
> Thank you for your comment. We will add some discussion in the paper about this issue.
>
> In the permuted MINST dataset, as you note, EWC does not have to know the task boundaries.
>
> However, consider disjoint MINST tasks, where you want to sequentially train task1 with handwritten digits 0,1,2, then task2 with 4,5,6, and then task3 with 7,8,9.
>
> For EWC, in the shared outputs case (the network only has 3 outputs), if you do not know the task ID during testing, then the network's output is ambiguous. For example, if, after softmax, you see that output 1 is the most probable one, you cannot tell whether this test sample is digit 0, or digit 4, or digit 7 (since these three digits share output 1). So you have to know the task ID during testing in the shared outputs case, to disambiguate which digit output 1 corresponds to. As you can see from our figure 3 a) and b), the EWC algorithm, with the knowledge of task ID, still fails to chance level after 20 epochs. So this is a fair comparison in our opinion.
>
> For the disjoint outputs case, you now have explicit unique representations for digits 0,1,2,4,5,6,7,8,9. Thus you do not need explicit task ID during test. But EWC's accuracy for task1 drops to 0% during testing after task2 and task3 have been trained, as it fails to map task1 test samples to the corresponding outputs. For example, after finishing training of task3 the network always outputs 7,8 or 9 for any test sample and has completely forgotten about 0,1,2,4,5,6; if you present a sample from the task1 test set, the output still is 7,8, or 9 only. Thus, in this case, we agree that EWC does not need to know the task ID while we do, but EWC does not work at all (0% correct on task1 and task2) while our method performs very well on all 3 tasks.

---

### Meta-Review · Area_Chair1 · 2018-12-13

**Confidence:** 5
**Recommendation:** Reject

**Metareview:**

The authors propose an approach for continual learning of a sequence of tasks which augments the network with task-specific neurons which encode 'adversarial subspaces' and prevent interference and forgetting when new tasks are being learnt. The approach is novel and seems to work relatively well on a simple sequence of MNIST or CIFAR10 classes, and has certain advantages, such as not requiring any stored data. However, the reviewers agreed that the presentation of the method is quite confusing and that the paper does not provide adequate intuition, visualisation, or explanation of the claim that they are preventing forgetting through the intersection of adversarial subspaces. Moreover, there was a concern that the baselines were not strong enough to validate the approach.